# Electron Transfer Induced Decomposition in Potassium–Nitroimidazoles Collisions: An Experimental and Theoretical Work

**DOI:** 10.3390/ijms20246170

**Published:** 2019-12-06

**Authors:** Mónica Mendes, Gustavo García, Marie-Christine Bacchus-Montabonel, Paulo Limão-Vieira

**Affiliations:** 1Atomic and Molecular Collisions Laboratory, CEFITEC, Department of Physics, Universidade NOVA de Lisboa, Campus de Caparica, 2829-516 Caparica, Portugal; mf.mendes@fct.unl.pt; 2Instituto de Física Fundamental, Consejo Superior de Investigaciones Científicas, Serrano 113-bis, 28006 Madrid, Spain; g.garcia@csic.es; 3Univ. Lyon, Université Claude Bernard Lyon 1, CNRS, Institut Lumière Matière, 69622 Villeurbanne, France

**Keywords:** nitroimidazoles, electron transfer, time-of-flight (TOF) mass spectrometry, fragmentation pattern, negative ions

## Abstract

Electron transfer induced decomposition mechanism of nitroimidazole and a selection of analogue molecules in collisions with neutral potassium (K) atoms from 10 to 1000 eV have been thoroughly investigated. In this laboratory collision regime, the formation of negative ions was time-of-flight mass analyzed and the fragmentation patterns and branching ratios have been obtained. The most abundant anions have been assigned to the parent molecule and the nitrogen oxide anion (NO_2_^–^) and the electron transfer mechanisms are comprehensively discussed. This work focuses on the analysis of all fragment anions produced and it is complementary of our recent work on selective hydrogen loss from the transient negative ions produced in these collisions. Ab initio theoretical calculations were performed for 4-nitroimidazole (4NI), 2-nitroimidazole (2NI), 1-methyl-4- (Me4NI) and 1-methyl-5-nitroimidazole (Me5NI), and imidazole (IMI) in the presence of a potassium atom and provided a strong basis for the assignment of the lowest unoccupied molecular orbitals accessed in the collision process.

## 1. Introduction

The chemical compounds 4-nitroimidazole (4NI, C_3_H_3_N_3_O_2_, 113 amu), 2-nitroimidazole (2NI, C_3_H_3_N_3_O_2_, 113 amu), 1-methyl-4-nitroimidazole (Me4NI, C_4_H_5_N_3_O_2_, 127 amu) and 1-methyl-5-nitroimidazole (Me5NI, C_4_H_5_N_3_O_2_, 127 amu) are nitro group containing molecules belonging to the large family of nitroimidazoles, with their molecular structures schematically shown in Figure 1. The interest in studying this class of molecules is increasing within the international scientific community especially because of their electron-affinic properties to be used as radiosensitisers in radiation treatments, namely in solid tumours growing in hypoxic environments [1]. These molecules are involved in several physical and chemical reactions inside the hypoxic cells where the nitroimidazole ring acts as a reducing agent leading to formation of radical anions. Moreover, nitroimidazoles are being investigated as novel compounds to act as oxygen, mimics, as well as inhibitors of carbonic anhydrase and new diagnostic imaging probes for hypoxic tumours [2]. In particular, nimorazole, has been investigated in several Danish clinical trials for the treatment of head and neck cancers showing promising results [3,4], specifically for patients with high concentrations of osteopontin in their plasma. The molecular mechanisms related to the reactions involving nitroimidazoles after irradiation are not fully understood yet. Therefore, comprehensive investigation of the underlying molecular mechanism that governs specific reactions should be considered in the investigation of new radiotherapeutic drugs and treatments. Feketeová et al. [5] reported on the formation of radical anions from radiosensitisers using electrospray ionization showing that the mechanism in misonidazole and nimorazole was closely related to the electron affinities of such molecules. Misonidazole shows a very different fragmentation pattern as compared with nimorazole which can be related to the observed higher toxicity in comparison to nimorazole. Pandeti et al. [6] have also investigated the nitroimidazolic radiosensitisers by electrospray ionization time-of-flight mass spectrometry and density functional theory. The electronic structure of nimorazole [7] and the formation of negative ions via dissociative electron attachment (DEA) processes were also investigated using different spectroscopic techniques. The recent work of Meißner et al. [8] has demonstrated that low-energy electrons are responsible for reduction of nimorazole via associative electron attachment, which corroborates the rationale that the radiosensitization process of this molecule in tumour cells is a result of reduction inside the cells. In contrast, Meißner and co-workers [9] have also investigated the role of low-energy electrons in misonidazole, showing that this molecule does not undergo reduction but rather dissociates in several anionic species leading to formation of radicals which are highly reactive to the hypoxic cell. Tanzer and co-workers [10,11] showed that low-energy electrons (0 to 8 eV) successfully decompose 4-nitroimidazole and two methylated isomers via DEA pointing out the significant importance of the study on the molecular mechanisms involved in these reactions (including detailed knowledge on the electronic and molecular structure) and emphasises the implications of that in tumour radiation therapy. Yu and Bernstein [12] reported by nanosecond laser excitation wavelength experiments that NO is generated by the decomposition of three distinct nitroimidazole molecules. They also have shown that vibrational warm and rotational cold distributions of the NO product are independent of the excitation wavelengths. Cartoni et al. [13] have studied VUV ionization induced decomposition of 2- and 4(5)-nitroimidazole by experimental and theoretical methods. Theoretical fixed-nuclei scattering calculations using the Schwinger multichannel (SMC) method were performed by Kossoski and Varella [14] to interpret the role of methylation at the N_1_ site of nitroimidazoles. 

Imidazole (IMI, 68 amu), C_3_H_4_N_2_ (Figure 1), is a five-membered aromatic molecule containing two nitrogen atoms and is polar in nature with a dipole moment of 3.8 D. This ring is found in several biological molecules, such as histidine, purines, nitroimidazole, antifungal drugs, and antibiotics [15]. The imidazole is also being used in the synthesis of new drugs and it is found in diverse molecules acting in inflammatory and analgesic mechanisms, as well as in carcinogenesis activities [16]. At the atomic and molecular level some studies have been performed regarding the integrity of the IMI ring. In DEA experiments, Ribar et al. [17] demonstrated that two low-energy shape resonances at 1.52 and 2.29 eV lead to the dehydrogenated anion (IMI−H)^−^ with the loss of an H atom stemming from the N_1_ position. These authors have also shown that upon electron attachment several more complex reactions involve the entire molecule, producing several small fragments associated to a complete dehydrogenation induced by a simple reaction. Modelli and Burrow [18] investigated by electron transmission spectroscopy, electron attachment energies of selected aza-derivatives including imidazole. They observed that replacement of a CH group in a ring with a nitrogen atom increases the electron-acceptor properties. Additionally, they also found experimentally the value of 3.12 eV for the imidazole vertical attachment energy (VAE), and two σ* orbitals (σ*_ring_ and σ*_NH_) at 7.6 and 5.9 eV. The excess electron binding energy of IMI was obtained by Rydberg electron transfer spectroscopy by Carles et al. [19]. Gianola et al. [20] investigated via negative ion photoelectron spectroscopy the imidazolide anion, which corresponds to the dehydrogenated parent anion, and obtained the electron affinities of the imidazolyl radicals upon deprotonation at different sites.

Here, we present negative ion formation in neutral potassium-neutral nitroimidazoles/imidazole collisions, together with ab initio calculations to support the experimental findings. In the next sections we briefly describe the results obtained, the experimental setup, the theoretical methods, and the main conclusions that can be drawn from this investigation. 

## 2. Results and Discussion 

In the gas phase, 4-nitroimidazole (4NI) and 5-nitroimidazole (5NI) concur in a tautomeric equilibrium [21,22] with DFT (Density Functional Theory) calculations showing a relative population of 1:0.7 for 4NI and 5NI at 390 K [7]. Thus, 4NI and 5NI are, from now on, referred to as 4(5)NI. In this section, we present experimental results on negative ion formation in electron transfer from neutral K atoms to a set of molecules, 4(5)NI, 2NI, Me4NI, Me5NI, and IMI, and theoretical methods to analyze the fragmentation pattern. Dissociative electron transfer TOF mass spectra were obtained at laboratory frame collision energies of 10 to 1000 eV (2.4 to 684 eV in the center of mass frame and from now on referred as available energy). Table 1 lists the assignments for the anions detected for the five molecules investigated.

Figure 2, Figure 3 and Figure 4 depict the TOF mass spectra recorded at 30, 100, and 500 eV lab frame collision energy for 4(5)NI, 2NI, Me4NI, and Me5NI, respectively. The branching ratios (BRs) for the major fragments of 4(5)NI, Me4NI, Me5NI are presented in Figure 5, Figure 6 and Figure 7, respectively. Figure 8 and Figure 9 show the TOF mass spectra at 30, 100, and 500 eV lab frame and the BRs for imidazole. 

The TOF mass spectra yields of 4(5)NI and 2NI are very similar and reveal that they are dominated by the parent anion (4(5)NI^−^ and 2NI^−^), NO_2_^−^, CN^−^, and the loss of OH. The methylated compounds also show a strong evidence of parent anion (Me4NI^−^ and Me5NI^−^), NO_2_^−^, and CN^−^ formation, whereas the loss of OH is completely quenched, as discussed recently [23]. Regarding the simple ring, imidazole, no parent anion is observed which can be rationalized in terms of the high electron affinity of the imidazolyl radical (IMI−H)^•^, 2.61 eV for the H loss in N_1_ position, and 1.99 eV for the H loss in C_5_ position [20]. Moreover, in electron transfer experiments with imidazole, the major anionic fragments are the dehydrogenated parent anion (IMI−H)^−^, CN^−^, and C_2_H_2_N^−^, the latter related to the loss of an HCN unit from the dehydrogenated parent anion, which also leads to CH^−^ formation (loss of another HCN unit). A similar HCN loss mechanism is also operative in more complex molecules and was already observed for potassium collisions with adenine and its derivative compounds [24] and pyrimidine [25]. The calculated highest occupied and lowest unoccupied molecular orbitals in Table 2 show a similar behaviour between 2NI and imidazole with molecular orbitals (MOs) delocalized on the ring and on the C‒H bonds. However, in the case of 4(5)NI and Me4NI/Me5NI the electron spin densities are mainly localized on the C=C bonds, while the presence of a ‒CH_3_ group in the methylated compounds relative to 4(5)NI, does not seem to change that much the shape of the MOs.

One of the most important differences found in the present results as compared with electron attachment measurements [11] is parent anion formation for all molecules investigated, with the exception of imidazole. We believe that the main reason for this difference resides on the presence of a potassium cation (K^+^) in the vicinity of the temporary negative ion (TNI) of the neutral molecule (M) formed upon electron transfer, i.e., K + M → (K^+^ M^−#^). In atom–molecule collisions, the electron transfer process happens when electrons follow adiabatically the nuclear motion in the vicinity of the crossing of the covalent and the ionic diabatic states [26] (see Figure 10). If we take as an example a diatomic molecule, the ionic potential energy curve lies above the covalent where the endoergicity (∆E) at large atom-molecule distances is given by ∆E = IE(K) − EA(AB), being IE the ionization energy of the potassium atom and EA the electron affinity of the target molecule. The strong Coulomb interaction of (K^+^ M^−#^) may delay autodetachment leading to a “stabilization” of the TNI, which can result in energy redistribution through the different available degrees of freedom leading either to a stable parent anion or different fragmentation channels. 

### 2.1. Nitroimidazoles and Methylated Compounds

#### 2.1.1. Parent Anion Formation

The parent anion formation from all nitroimidazole molecules is one of the most remarkable processes observed in these experiments, especially for high impact collision energies. The BRs as a function of the available energy (Figure 5, Figure 6 and Figure 7) show that the parent anion is formed for all energies and, in the case of 4(5)NI, represents the most intense anion yield between 5.7 eV and 664 eV available energy. At higher collision energies, 4(5)NI^−^ accounts for ~60% of the total anion yield, whereas methylation at the N_1_ position in Me4NI and Me5NI enhances parent anion formation up to ~70% and ~60% of total anion yield. However, for Me5NI at energies between 6 and 13 eV the most intense fragment is (Me5NI–CH_3_)^−^, and for Me4NI at energies between 16 and 51 eV the most intense anion is NO_2_^−^. 

The electron spin densities of Table 2 reveal that in the LUMOs of the nitroimidazole compounds, electron promotion may be localized in the five-membered ring. However, due to nitrogen dioxide high electron affinity (2.2730 ± 0.0050 eV) [28] a reasonable delocalization of the LUMOs is expected over NO_2_, which is not visible due to the truncation methodology employed to plot the MOs. Thus, parent anion formation can only be rationalized in terms of strong competition between the ring and NO_2_ upon electron transfer. Actually, the branching rations in Figure 5, Figure 6 and Figure 7 clearly show either the parent anion or NO_2_^−^ formation, processes which compete as a function of the collision energy. The yield of these anions is also dependent on the collision time and the efficient mechanism of intramolecular electron transfer from the ring to NO_2_ and vice-versa, which may be related with the stabilizing effect of K^+^ in the vicinity of the TNI. Additionally, the calculated LUMOs in Table 2 also show the different character of 4(5)NI, 1Me4NI and Me5NI, and σ*(C=C), relative to the π*(ring) of 2NI. Given that in principle an electron promotion to a π* orbital does not lead to bond breaking, unless a σ* is diabatically crossed with that MO, parent anion formation relative to NO_2_^‒^ in 2NI is expected to be more enhanced than in 4(5)NI, Me4NI, and Me5NI. This behavior is clearly visible in the relative intensities of the anions from the TOF mass spectra of Figure 2. In this collision energy regime (~16 eV available energy), accessing the lowest MOs seems plausible, for example in 2NI the π* ring at 4.9 eV, whereas at intermediate energies (see Figure 3) accessing higher energy π* orbitals (6.1 and/or 7.4 eV) is possible as well as an efficient crossing with σ* antibonding orbital (9.0 eV). Then, this will yield an enhanced fragmentation pattern where the ratio between parent anion and NO_2_^‒^ is lower, as is the case of 2-nitroimidazole. 

In DEA studies, the parent anion formation is not observed in 4(5)NI, but in Me4NI and Me5NI [10,11]. Tanzer et al. have shown that the parent anion in Me4NI and Me5NI are formed within a resonance close to 0 eV. This was explained based on the assumption that the energy involved in the electron-molecule system, consisting of the energy of the incoming electron and the electron affinity of the molecule, is transferred to the vibrational degrees of freedom of the TNI, which delays autodetachment, resulting in a long-lived TNI in the μs time scale detection window. These molecules also have positive electron affinities which means that the ground state of the anion lies below the neutral molecule. Molecules with positive electron affinities lead to formation of stable anions where the extra electron is in a bound state [29]. Following this rationale, observation of the parent anion for all molecules is not surprising. 

However, for higher collision energies, which means fast collisions, the presence of the parent anion as the most intense ion seems not to follow completely the so-called adiabatic principle. Briefly, the adiabatic principle describes that if the internuclear distance in a diatomic molecule changes rapidly (fast collisions), the electron transfer cannot occur. In that case, the system remains in the same electronic configuration or diabatic state [30]. In the low-energy collision regime, the electron is transferred from the potassium atom to the nitroimidazole molecular target during the approach, i.e., at the first crossing of the potential energy curves involved in the collision process (see Figure 10). Given the positive electron affinity of the target, a TNI can be formed where the excess of internal energy can be distributed over the different degrees of freedom delaying autodetachment. In this regime, with typical collision times of the order of several tens of fs, the presence of the K^+^ ion in the vicinity of the TNI (strong Coulomb interaction) can also allow intramolecular electron transfer yielding dissociative channels, in particular fragments with high electron affinities as NO_2_ and CN, with the resulting anions with a lifetime long enough to be detected through mass spectrometry. This description is much more attuned to an adiabatic process and this rationale seems reasonable since in the low-energy regime, typically below 50 eV, we observe the highest yields of fragment anions. However, when the collision energy is increased, the potassium atom may not transfer its electron at the first crossing but rather at the second (see Figure 11). In this case, as the K^+^ ion moves away from the TNI, there is no time enough for efficient intramolecular electron transfer and a nonadiabatic description is favourable, i.e., diabatic states can be described as those where the electronic character does not change with respect to the nuclear coordinates. This makes sense since at high-collision energies, typically collision times of the order of few fs, some vibrational modes within the molecular frame can be considered frozen [31].

In electron attachment studies, Feketeová et al. [5] and Meißner et al. [8] have also observed that the parent anion radical (M^●−^) has the highest yield in the nimorazole molecule, a derivative of 5-nitroimidazole. Additionally, Meißner et al. showed that DEA processes are supressed in favour of associative attachment when nimorazole is solvated in water, and the parent anion formation is stabilized due to energy dissipation to the surrounding environment. These results reinforce the effect that a third body (K^+^) in the collisional system may stabilize the parent anion of these molecules. These results strongly suggest that radical anion formation seems to be the initial step for the impact of nitroimidazoles as radiosensitisers in hypoxic tumour cells [8]. 

#### 2.1.2. Formation of NO_2_^−^, Loss of Neutral NO_2_, and Related Multiple Dehydrogenation (HNO_2_^−^, H_2_NO_2_^−^, and H_3_NO_2_^−^)

The NO_2_^−^ yield is, together with the parent anion, one of the most intense in the nitroimidazoles TOF mass spectra, which is unsurprising because of NO_2_ electron affinity, 2.27 eV [28]. For lower collision energies (e.g., 30 eV in the lab frame, Figure 2) and in the case of the methylated compounds, NO_2_^−^ corresponds to the most intense anion. Formation of NO_2_^−^ occurs after cleavage of the simple C−NO_2_ bond, Reaction (1). However, bond breaking may also lead to the complementary reaction yielding (M−NO_2_)^−^, Reaction (2).
(1)$$K+M\to \left(K^{+}M^{-\#}\right)\to K^{+}+{\left(M\right)}^{-\#}\to K^{+}+{\mathrm{NO}}_{2}^{-}+\left(M-{\mathrm{NO}}_{2}\right)$$
(2)$$K+M\to \left(K^{+}M^{-\#}\right)\to K^{+}+{\left(M\right)}^{-\#}\to K^{+}+{\left(M-{\mathrm{NO}}_{2}\right)}^{-}+{\;\mathrm{NO}}_{2}$$

The loss of a neutral NO_2_ leading to (M−NO_2_)^−^ formation is also observed for all nitroimidazole molecules studied although barely visible in 4(5)NI. However, the intensity of the fragments is much lower than the complementary reaction yielding NO_2_^−^. These results are in quite good agreement with DEA studies [10,11], where for methylated compounds, below 2 eV, Reaction (2) is completely suppressed, meaning that methylation blocks the C_4_ and C_5_ positions. This is not observed from the present results where the appearance energy of (M−NO_2_)^−^ occurs at approximately the same energy as in 4(5)NI, 2NI, and methylated molecules (around 30 eV lab frame). From the calculated MOs in Table 2, generally speaking, transitions from the HOMOs to the LUMOs reveal that the overlap is more efficient in the case of 4(5)NI, Me4NI, and Me5NI rather than in 2NI, which means an enhanced yield of NO_2_^−^ for the former set of molecules than the latter.

Other set of reactions related to nitroimidazoles is the formation of anionic fragments at 66, 65, and 64 *m*/*z*, which are assigned to (M−HNO_2_)^−^, (M−H_2_NO_2_)^−^ and (M−H_3_NO_2_)^−^. These ions are only observed for 4(5)NI and 2NI but have not been reported in DEA studies at low-electron impact energies [11]. The anions’ formation in potassium collisions with 4(5)NI/2NI may result from the loss of a neutral H atom from the N_1_ site together with the loss of NO_2_ resulting in (M−HNO_2_)^−^ formation. This is different from the DEA experiments [11] where at low energies and upon methylation, this anionic channel is completely quenched. 

#### 2.1.3. (M–NO)^‒^ and (M–OH)^‒^ Formation 

The loss of a neutral OH^⦁^ (96 *m*/*z*) radical is one of the most intense and important anionic fragments observed in collisions of K atoms with 4(5)NI and 2NI (see Figure 2, Figure 3 and Figure 4). Furthermore, after methylation at the N_1_ site, the channel that leads to formation of the anion is completely blocked, as thoroughly discussed in reference [23]. A similar process seems to occur in (M−NO)^−^ formation, although the yield is much lower. These two anions can be formed through the following reactions: (3)$$K+M\to \left(K^{+}M^{-\#}\right)\to K^{+}+{\left(M\right)}^{-\#}\to K^{+}+{\left(M-\mathrm{OH}\right)}^{-}+{\;\mathrm{OH}}^{●}$$
(4)$$K+M\to \left(K^{+}M^{-\#}\right)\to K^{+}+{\left(M\right)}^{-\#}\to K^{+}+{\left(M-\mathrm{NO}\right)}^{-}+{\;\mathrm{NO}}^{●}$$

The formation of these two negative ions requires the cleavage of several bonds, namely C−N, C−H, and N−O. Since the loss of neutral OH^⦁^ is suppressed upon methylation, we conclude that the H atom involved is originating from the N_1_ position. The threshold of (M−OH)^−^ formation is 5.7 eV (15 eV lab frame) and 2.5 eV (10 eV lab frame), for 4(5)NI and 2NI, respectively. Although the BR for (M−NO)^−^ is not shown here, the threshold in the case of 4(5)NI (19.8 eV) is much higher than in 2NI (5.7 eV). These results are consistent with the three DEA resonances found below 3 eV [11,32] where 2NI shows an enhancement in the fragmentation pattern as compared with 4(5)NI. These two reactions are quite important within the biological environment further to low-energy electron transfer collisions. Formation of highly reactive species such as OH^•^ and NO^•^ in the cellular environment, induces a set of reactions resulting in damage of the biocomponents including the DNA molecule [33,34,35,36]. Since nitroimidazoles are being investigated as attuned radiosensitisers in radiotherapy of hypoxic tumours, the presence of the reaction channels may be particularly useful in the improvement of dedicated radiation protocols. 

#### 2.1.4. CN^−^ Formation

The TOF mass spectra in Figure 2, Figure 3 and Figure 4 show that CN^−^ also contributes significantly to the total anion pattern of 4(5)NI, 2NI, Me4NI and Me5NI. The cyanide anion can be formed through the loss of an N atom from the NO_2_ group or by excision of a CN unit directly from the ring. Both processes are formed after a complex rearrangement of the transient negative ion. The cyanide anion was already observed in other biomolecules, such as pyrimidine [25], adenine [24], thymine and uracil [37] upon electron transfer with potassium as electron donor. According to the BR results (see Figure 5, Figure 6 and Figure 7), the CN^−^ threshold of formation is 9 eV for all molecules except for 4(5)NI, which is below 2.4 eV. This agrees well with DEA experiments where resonances near 0 and 4 eV for 4(5)NI and 2NI, and around 4 eV in the case of methylated molecules [11,32] were observed.

The cyanide anion yields, as shown in Figure 2, are discernible, although with modest intensity, for 2NI, 4(5)NI, and Me4NI, whereas in the case of Me5NI it is completely suppressed. The calculated MOs, shown in Table 2, show similar electron spin densities. However, at intermediate collision energies (Figure 3), CN^−^ formation is possible for all investigated molecules but is more enhanced in the case of 2NI. This is certainly expected from the considerable π* ring character of the LUMO at 4.9 eV and even at 6.1 eV, that can be accessed at these energies. 

### 2.2. Imidazole Molecule

#### 2.2.1. (IMI−H)^−^ Formation

The most intense anionic fragment detected in K-imidazole collisions in the wide energy range (15 to 1000 eV) investigated, is the dehydrogenated parent anion, (IMI−H)^−^ (Figure 8 and Figure 9). The BR for (IMI−H)^−^ shows that it decreases from 5.4 eV (17 eV lab frame) to 52.8 eV (100 eV lab frame) where it reaches a plateau contributing to ~40% of the total anion yield. In Figure 9 it is also noticeable that for other fragments such as CN^−^ and C_2_H_2_N^−^ their BRs increase in the energy region where the (IMI−H)^−^ yield decreases. This demonstrates that the hydrogenated parent anion may be a precursor in the formation of other anions. This behavior was already shown in electron transfer collisions in neutral potassium atoms with adenine [24], a molecule that is formed by a pyrimidine and an imidazole ring. Formation of (IMI−H)^−^ can occur from the cleavage of an C−H or N−H bond, represented as:(5)$$K+\mathrm{IMI}\to \left(K^{+}{\mathrm{IMI}}^{-\#}\right)\to K^{+}+{\left(\mathrm{IMI}\right)}^{-\#}\to K^{+}+{\left(\mathrm{IMI}-H\right)}^{-}+\;H$$

From the BRs (see Figure 9) the appearance threshold of (IMI−H)^−^ is at around 5.4 eV (17 eV lab frame). The electron affinity of the imidazolyl radical (IMI−H)^•^ was determined through negative ion photoelectron spectroscopy to be 2.613 eV in the case of deprotonation at the N_1_ position, and 1.992 eV in the case of deprotonation at the C_5_ position [20]. Cunha et al. [38] showed in adenine and derivative molecules that the dehydrogenated parent anion is formed from the loss of an H atom from the N_9_ site which corresponds to the N_1_ position in the imidazole moiety. Moreover, DEA measurements [17] show two shape resonances at 1.52 eV and 2.29 eV which are related to the formation of this anion through the loss of an H atom at the N_1_ position. Two other resonances have been reported for higher energies (~7 and 11 eV) and assigned to core excited resonances. In Table 2 we show the four lowest calculated π* MOs at 5.2 eV (π*ring), 7.2 eV (π*C=C), and 8.3 eV (π*C‒H) and a σ* resonance (C‒H) at 10.1 eV. Because in electron transfer processes an electronic transition accessing π* states does not lead to direct bond cleavage unless a repulsive σ* is crossed diabatically, we may assume that the loss of a neutral H at low-collision energies can proceed through a coupling between the π* state at 5.2 eV (ring) or 7.3 eV (C=C) and the dissociative σ* state (C‒H) localized at 10.1 eV. Alternatively, for higher collision energies a direct initial electron transfer to the σ* state and subsequent dissociation can take place, which may explain (IMI−H)^−^ higher yields at higher collision energies (e.g., σ* CH at 10.1 eV, see Table 2). 

#### 2.2.2. C_2_H_2_N^−^, CH^−^ and CN^−^ Formation

Formation of C_2_H_2_N^−^ can proceed from the loss of an HCN unit from the dehydrogenated parent anion, whereas CH^−^ may result from the loss of another HCN unit from C_2_H_2_N^−^. This behavior was also previously observed in the case of K‒adenine collisions with the HCN loss mechanism discussed in detail [24]. The BRs in Figure 9 show that C_2_H_2_N^−^ and CH^−^ thresholds of formation are 10 eV (25 eV lab frame) and 24.3 eV (50 eV lab frame), respectively. In DEA experiments C_2_H_2_N^−^ formation possesses two resonances at 6.6 and 10.0 eV [17]. 

Figure 8 and Figure 9 show CN^−^ as one of the most relevant anions formed in K−imidazole collisions, which seems reasonable due to the electron affinity of the cyano radical (3.8620 ± 0.0050 eV) [28]. Indeed, it is the second most intense fragment anion between 6.5 and 35.7 eV (19 eV and 70 eV lab frame). The relative intensity of this anion to the dehydrogenated parent anion at ~13 eV can be explained by initial electron transfer to the lowest MO (see Table 2), that is mainly of π* ring character, and therefore not particularly efficient yielding CN^−^. We observe that CN^−^ threshold of formation is 6.5 eV (19 eV lab frame). For higher collision energies, especially for 167 eV (300 eV lab frame), we also observe a strong competition between CN^−^ and C_2_H_2_N^−^ formation. The electron affinity of the cyanomethyl radical is (1.530 ± 0.010 eV) [28] which may indicate that the imidazole collision induced dissociation process is mainly dictated (>53 eV, see Figure 8) by the excess energy that is deposited in IMI through the available internal degrees of freedom. Similarly to the mechanism described in K collisions with adenine [24], uracil, and thymine [37], and in contrast to K−pyrimidine studies [25], in imidazole, CN^−^ formation may also proceed from the decomposition of the dehydrogenated parent anion.

## 3. Experimental Methods 

The crossed molecular beam setup used to study collisions of neutral potassium (K) atoms with neutral nitroimidazoles, imidazole, and related molecules, has been described in detail elsewhere [23,39]. Briefly, an effusive target molecular beam crosses a primary beam of fast neutral K atoms and the product anions are analyzed using either a reflectron (KORE R-500-6) or a dual-stage linear time-of-flight (TOF) mass spectrometers. The K beam is produced in a resonant charge exchange chamber from the interaction of K^+^ ions from a potassium ion source (10 to 1000 eV in the lab frame) with gas-phase neutral potassium atoms from an oven source. The TOF anion yield is normalized considering the primary beam current, pressure and acquisition time. Negative ions formed in the collision region were extracted by a ~380 V/cm pulsed electrostatic field. The typical base pressure in the collision chamber was 6 × 10^−5^ Pa and the working pressure was 2 × 10^−4^ Pa. Mass spectra (resolution m/Δm ≈ 800 and ≈ 125 for the reflectron and linear TOF) were obtained by subtracting background measurements (without the sample) from the sample measurements. Mass calibration was performed on the basis of the well-known anionic species formed after potassium collisions with nitromethane [39]. The molecular samples 4-nitroimidazole, 2-nitroimidazole and 1-methyl-5-nitroimidazole were supplied by Sigma-Aldrich with a stated purity of ≥97%. The 1-methy-4-nitroimdiazole sample was purchase by Fluorochem with a stated purity of ≥95%, and imidazole was supplied by Alpha Aesar with a stated purity of 99%. The solid samples were used as delivered. 

## 4. Theoretical Method

The charge transfer process of a neutral potassium atom with a selected biomolecule is described theoretically in the framework of the molecular representation of the collisions considering the evolution of the quasi-molecular system formed by the potassium projectile and the molecular target along the reaction coordinate. The calculations are performed in the one-dimension reaction coordinate approximation implemented with success in previous ion/neutral-biomolecule collision systems [40,41,42] where the collision system is treated as a pseudo-diatomic molecule evolving along the coordinate associated with the distance between the projectile and the biomolecule [43,44]. This approach does not take into account the internal degrees of freedom of the biomolecular target but remains relevant in very fast collision processes where nuclear vibrational and rotational motions are much slower than the collision time [45], and thus can be considered frozen during the process. The geometries of imidazole compounds have been optimized at the MP2 level of theory from the work of Calvo et al. [46] (see Supplementary Materials). The potassium atom has been assumed to collide perpendicularly to the biomolecule, pointing at the center of the imidazole heterocycle, as the charge transfer process has been clearly shown to be favoured in such orientation for pyrimidine targets [45,47]. Calculations have been performed by means of ab initio methods with the MOLPRO code [48] keeping imidazole targets frozen in their ground state geometry in all cases and using Cartesian coordinates, with no symmetries. All electrons have been taken into account for C, N, O, and H atoms with the VTZ (valence triple zeta) basis set, although the 18 core electrons of potassium have been treated through the ECP18sdf core-electron pseudopotential [49], with the corresponding basis set. The natural molecular orbitals of the imidazole compounds in the presence of potassium have been determined in the asymptotic region at the state-averaged CASSCF level of theory from a CAS (3,11) calculation. For this procedure we have taken the static electron correlation as in previous work [25] and involving three active electrons and eleven active orbitals including molecular orbitals on imidazole compounds and s,p orbitals on potassium. The resultant higher occupied molecular orbital (HOMO) and lowest unoccupied molecular orbitals (LUMOs) for imidazole, nitroimidazole and methyl-nitroimidazoles are shown in Table 2. Imidazole and 2-nitroimidazole exhibit quite similar behaviour, with a HOMO located mainly on the C=C bond of the heterocycle while the LUMOs are mostly π* and σ* orbitals delocalized on the heterocycle and on the C–H bond. The presence of a nitro group in position 4 or 5, on the carbon of the C=C bond, induces a polarization of the HOMO towards the C=N bond and thus drives mainly the LUMOs π* and σ* located on the C=C bond, which may be observed for the series 4-nitroimidazole and 1-methyl-4-nitroimidazole and 1-methyl-5-nitroimidazole.

## 5. Conclusions

The study gives a comprehensive insight on the decomposition mechanisms of neutral imidazole-related compounds (4(5)NI, 2NI, Me4NI, Me5NI, and IMI) in electron transfer collisions with neutral potassium atoms. In the case of nitroimidazoles (4(5)NI, 2NI, and methylated molecules) the parent anion was identified as the most intense ion for lab frame collision energies between 10 and 1000 eV, in contrast to DEA studies, where the parent anion is only observed in the case of methylated molecules for very low energies (near 0 eV). This difference may be rationalised in terms of the presence of a third body (K^+^) in the collisional complex system (K^+^ M^−#^). Other important anionic species were assigned as part of the fragmentation pattern of nitroimidazoles, such as NO_2_^−^, the loss of OH^⦁^ and NO^⦁^. Some differences were also found between the 4(5)NI and 2NI molecules, and between the methylated compounds, suggesting that the nitro group position influences the fragmentation pattern. Additionally, the studies with imidazole show that the NO_2_ group in the ring brings considerable modifications in the fragmentation pattern, as compared with the nitroimidazoles. In the imidazole molecule, the TOF mass spectra are dominated by the dehydrogenated parent anion, CN^−^ and C_2_H_2_N^−^. We have also observed that the dehydrogenated parent anion is a precursor in the formation of other fragment anions, which was also verified in other K-uracil, -thymine and -adenine electron transfer experiments. The decomposition channels presented here upon electron transfer showing formation of parent anion and other radicals even at higher energies, demonstrate the capability of these nitroimidazole compounds to act as radiosensitizers in medical treatments with radiation. Some differences found between nitroimidazole molecules and related methylated compounds as well as the differences observed in imidazole molecules demonstrate the importance of these studies in understanding the underlying molecular mechanisms that govern the electron induced decomposition which can be used to formulate new radiopharmaceutical compounds or new methodologies to make these chemical compounds more attuned to key specific reaction.

## Figures and Tables

**Figure 1 ijms-20-06170-f001:**
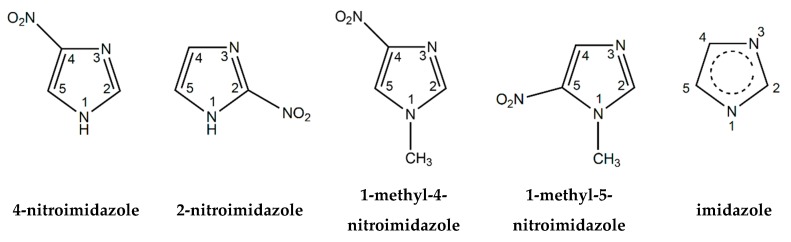
Molecular structure of 4-nitroimidazole (4NI), 2-nitroimidazole (2NI), 1-methyl-4-nitroimidazole (Me4NI), 1-methyl-5-nitroimidazole (Me5NI) and imidazole (IMI).

**Figure 2 ijms-20-06170-f002:**
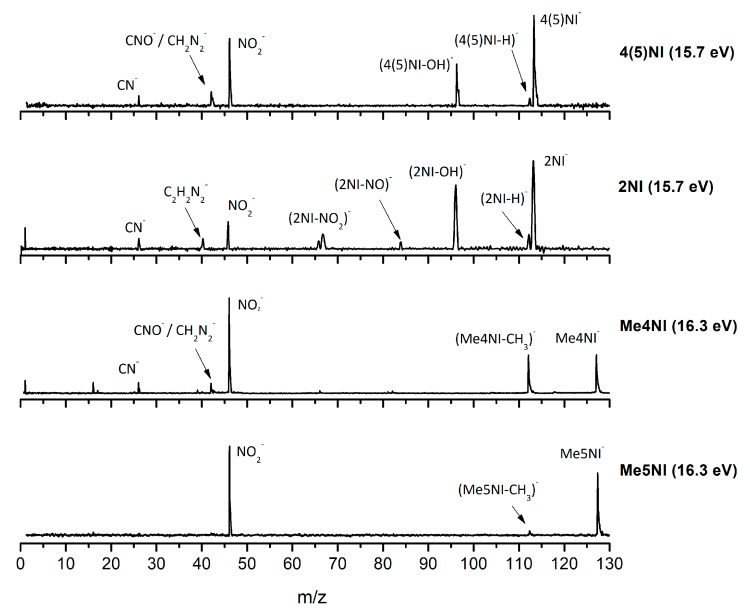
Time-of-flight negative ion mass spectra in potassium collisions with 4(5)-nitroimidazole (4(5)NI), 2-nitroimidazole (2NI), 1-methyl-4-nitroimidazole (Me4NI), and 1-methyl-5-nitroimidazole (Me5NI) at 30 eV lab frame energy (15.7 and 16.3 eV available energy in the center of mass, respectively). See text for details.

**Figure 3 ijms-20-06170-f003:**
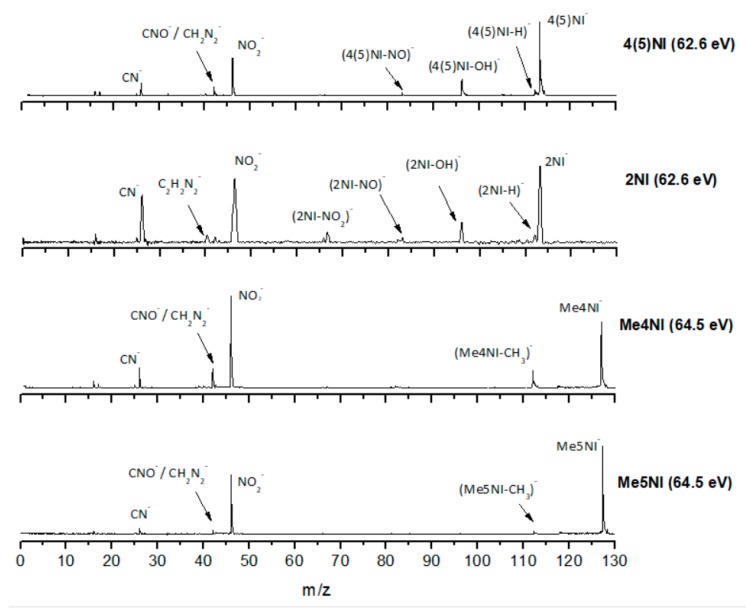
Time-of-flight negative ion mass spectra in potassium collisions with 4(5)-nitroimidazole (4(5)NI), 2-nitroimidazole (2NI), 1-methyl-4-nitroimidazole (Me4NI), and 1-methyl-5-nitroimidazole (Me5NI) at 100 eV lab frame energy (62.6 and 64.5 eV available energy in the center of mass, respectively). See text for details.

**Figure 4 ijms-20-06170-f004:**
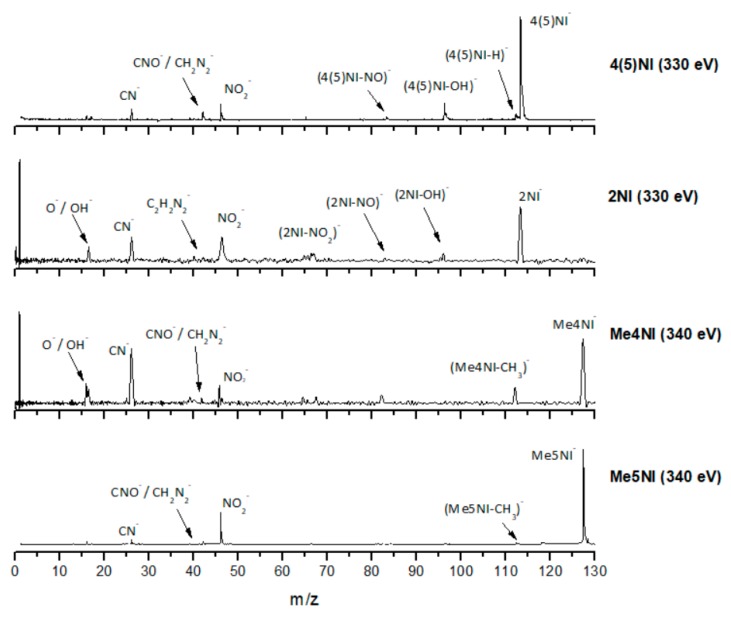
Time-of-flight negative ion mass spectra in potassium collisions with 4(5)-nitroimidazole (4(5)NI), 2-nitroimidazole (2NI), 1-methyl-4-nitroimidazole (Me4NI), and 1-methyl-5-nitroimidazole (Me5NI) at 500 eV lab frame (330 and 340 eV available energy in the center of mass, respectively). See text for details.

**Figure 5 ijms-20-06170-f005:**
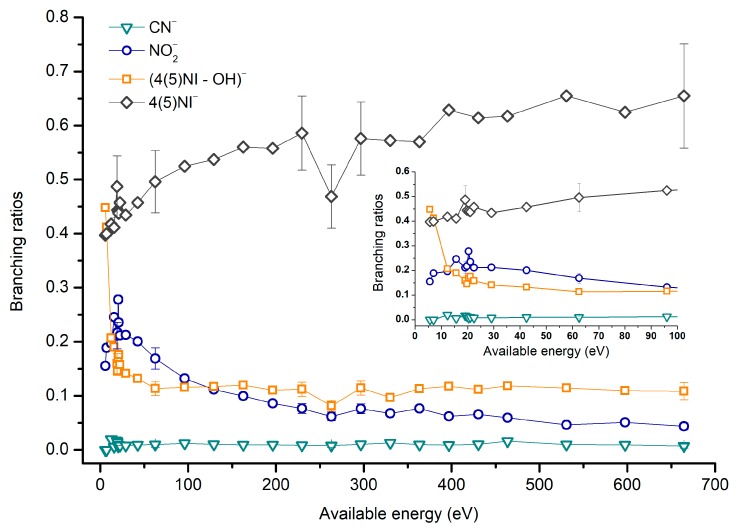
4(5)-Nitroimidazole (4(5)NI) branching ratios (fragment anion yield and total anion yield) of the main negative ions formed as a function of the collision energy in the center of mass frame. Error bars related to the experimental uncertainty associated with the ion yields have been added to a few data points in order to avoid congestion of the figure. The lines are just to guide the eye. See text for details.

**Figure 6 ijms-20-06170-f006:**
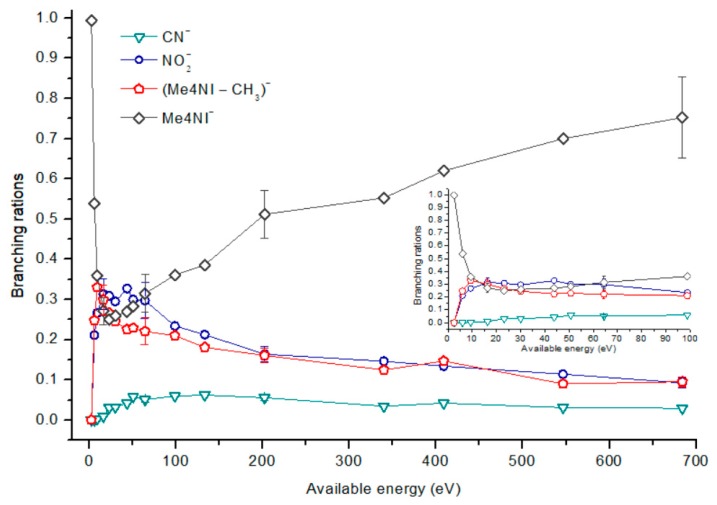
1-Methyl-4-nitroimidazole (Me4NI) branching ratios (fragment anion yield and total anion yield) of the main negative ions formed as a function of the collision energy in the center of mass frame. Error bars related to the experimental uncertainty associated with the ion yields have been added to a few data points in order to avoid congestion of the figure. The lines are just to guide the eye. See text for details.

**Figure 7 ijms-20-06170-f007:**
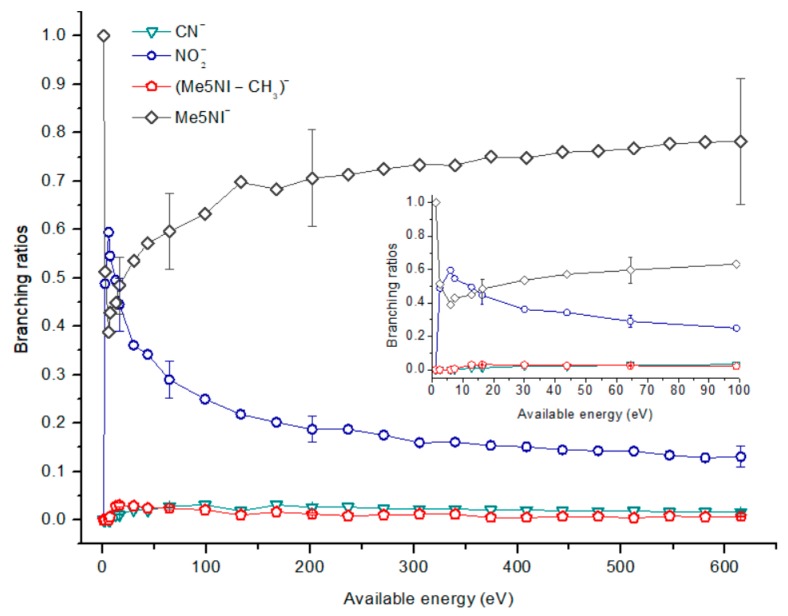
1-Methyl-5-nitroimidazole (Me5NI) branching ratios (fragment anion yield and total anion yield) of the main negative ions formed as a function of the collision energy in the center of mass frame. Error bars related to the experimental uncertainty associated with the ion yields have been added to a few data points in order to avoid congestion of the figure. The lines are just to guide the eye. See text for details.

**Figure 8 ijms-20-06170-f008:**
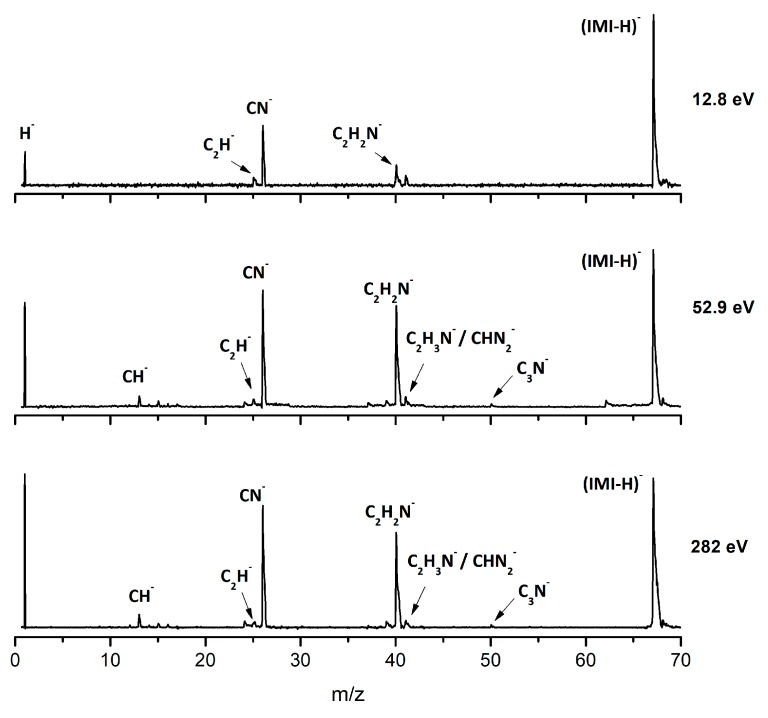
Time-of-flight negative ion mass spectra in potassium–imidazole (IMI) collisions at 30, 100 and 500 eV lab frame energy (12.8, 52.9 and 282 eV available energy in the center of mass, respectively). See text for details.

**Figure 9 ijms-20-06170-f009:**
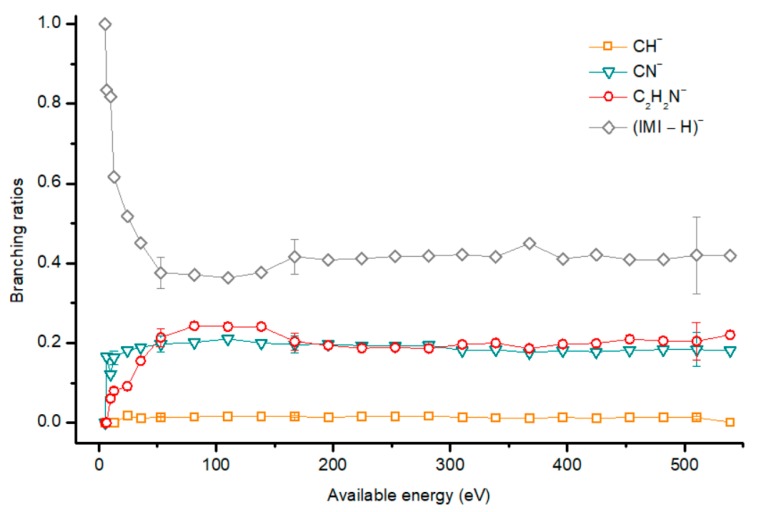
Imidazole (IMI) branching ratios (fragment anion yield/total anion yield) of the main negative ions formed as a function of the collision energy in the center of mass frame. Error bars related to the experimental uncertainty associated with the ion yields have been added to a few data points in order to avoid congestion of the figure. The lines are just to guide the eye. See text for details.

**Figure 10 ijms-20-06170-f010:**
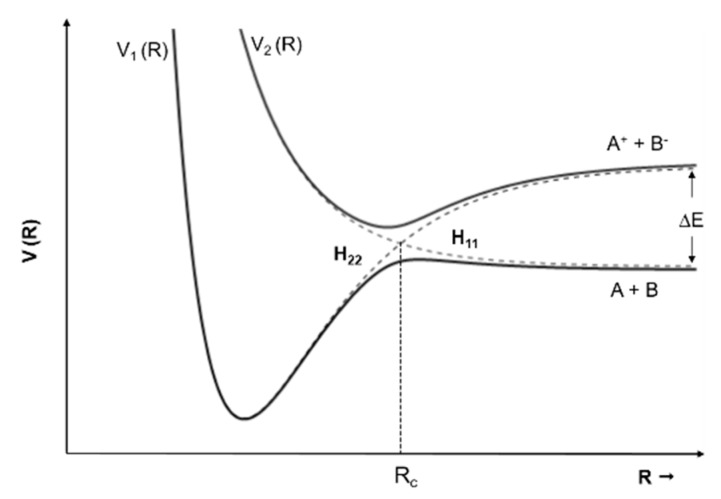
Schematics of adiabatic and diabatic potential energy curves for an atom–atom collision. V_1_(R) and V_2_(R) represent the adiabatic states (full curves). The dashed lines represent the diabatic states: H_11_ (covalent) and H_22_ (ionic). R_c_ is the crossing radius and ∆E is the endoergicity. Adapted from [27].

**Figure 11 ijms-20-06170-f011:**
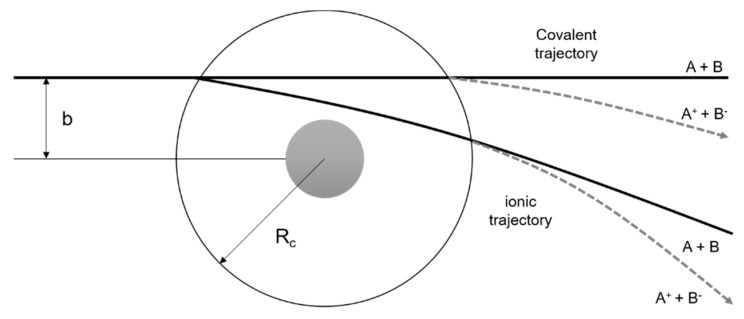
Schematics of atom-atom scattering representing the four possible pathways considering the impact parameter b of the incoming projectile and two different crossing radii. The dashed area represents the repulsive part of the potential. Adapted from [30].

**Table 1 ijms-20-06170-t001:** Assignment of the negative ions formed in potassium collisions with 4(5)-nitroimidazole (4(5)NI), 2-nitroimidazole (2NI), 1-methyl-4-nitroimidazole (Me4NI), 1-methyl-5-nitroimidazole (Me5NI), and imidazole (IMI).

*m*/*z*	4(5)NI	2NI	Me4NI	Me5NI	IMI
1	H^−^	H^−^	H^−^	H^−^	H^−^
12					C^−^
13				CH^−^	CH^−^
14					CH_2_^−^
15				CH_3_^−^/NH^−^	CH_3_^−^/NH^−^
16	O^−^/NH_2_^−^	O^−^/NH_2_^−^	O^−^/NH_2_^−^	O^−^/NH_2_^−^	O^−^/NH_2_^−^
17	OH^−^	OH^−^	OH^−^	OH^−^	
24			C_2_^−^	C_2_^−^	C_2_^−^
25	C_2_H^−^		C_2_H^−^	C_2_H^−^	C_2_H^−^
26	CN^−^	CN^−^	CN^−^	CN^−^	CN^−^
39	C_2_HN^−^/C_3_H_3_^−^		C_2_HN^−^/C_3_H_3_^−^	C_2_HN^−^/C_3_H_3_^−^	C_2_HN^−^/C_3_H_3_^−^
40	C_2_H_2_N^−^	C_2_H_2_N^−^	C_2_H_2_N^−^	C_2_H_2_N^−^	C_2_H_2_N^−^
41					C_2_H_3_N^−^/CHN^−^
42	CNO^−^/CH_2_N^−^	CNO^−^/CH_2_N^−^	CNO^−^/CH_2_N^−^	CNO^−^/CH_2_N^−^	
46	NO_2_^−^	NO_2_^−^	NO_2_^−^	NO_2_^−^	
50					C_3_N^−^
64	(4(5)NI–H_3_NO_2_)^−^				
65	(4(5)NI–H_2_NO_2_)^−^				
66	(4(5)NI–HNO_2_)^−^	(2NI–HNO_2_)^−^	(Me4NI–CH_3_NO_2_)^−^	(Me5NI–CH_3_NO_2_)^−^	
67	(4(5)NI–NO_2_)^−^	(2NI–NO_2_)^−^			(IMI–H)^−^
68					(IMI–H)^−^ *
81			(Me4NI–NO_2_)^−^	(Me5NI–NO_2_)^−^	
82	(4(5)NI–HNO)^−^		(Me4NI–CH_3_NO)^−^	(Me5NI–CH_3_NO)^−^	
83	(4(5)NI–NO)^−^	(2NI–NO)^−^			
96	(4(5)NI–OH)^−^	(2NI–OH)^−^	(Me4NI–HNO)^−^	(Me5NI–HNO)^−^	
112	(4(5)NI–H)^−^	(2NI–H)^−^	(Me4NI–CH_3_)^−^	(Me5NI–CH_3_)^−^	
113	4(5)NI^−^	2NI^−^			
114	4(5)NI^−^ *				
127			Me4NI^−^	Me5NI^−^	
128			Me4NI^−^ *	Me5NI^−^ *	

* means isotopic contribution.

**Table 2 ijms-20-06170-t002:** Calculated highest occupied molecular orbitals (HOMOs) and lowest unoccupied molecular orbitals (LUMOs) for potassium (K) + imidazole and nitroimidazole-related compounds.

IMI	2NI	4NI	Me4NI	Me5NI
HOMO π_C=C_−9.3 eV 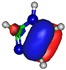	HOMO π_C=C_−10.4 eV 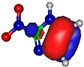	HOMO π_C=N_−11.8 eV 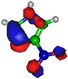	HOMO π_C=N_−11.4 eV 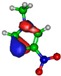	HOMO π_C=N/C=C_−11.4 eV 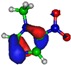
LUMO π*_ring_5.2 eV 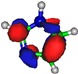	LUMO π*_ring_4.9 eV 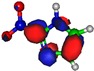	LUMO σ*_C=C_6.5 eV 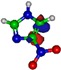	LUMO σ*_C=C_7.3 eV 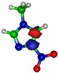	LUMO σ*_C=C_8.9 eV 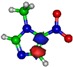
π*_C=C_7.2 eV 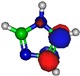	π*_C=C_6.1 eV 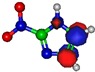	π*_C=C_10.9 eV 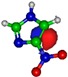	π*_C=C_11.3 eV 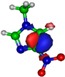	π*_C=C_11.7 eV 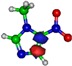
π*_CH_8.3 eV 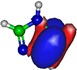	π*_CH_7.4 eV 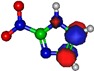	π*_C=C_13.2 eV 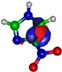	π*_C=C_14.9 eV 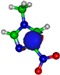	π*_C=C_15.4 eV 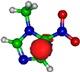
σ*_CH_10.1 eV 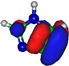	σ*_CH_9.0 eV 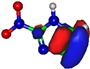

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
