# Peer review of "Electron Transfer Induced Decomposition in Potassium–Nitroimidazoles Collisions: An Experimental and Theoretical Work"

_ijms, 2019, doi:10.3390/ijms20246170_

Round 1

Reviewer 1 Report

The manuscript by Mendes et al. reports a study on the evolution and fragmentation of imidazole and nitroimidazole derivatives under the effects of potentially ionizing radiation. in particular the collision with neutral K atoms and the opening of anion base dissociation pathways is particularly focused on. This is particularly interested since such compounds are used as sensitizers in radiotherapy, and hence the proper rationalization of their inherent properties and the response to ionizing radiation is vital to the design of improved therapeutical protocols. 

One of the most important finding of this work being the dominant presence of the parent anion in mass spectroscopy. 

However, despite the interest and potential impact of the work I definitively feel that the manuscript and the presentation require significant major revision. 

In particular the role of molecular modeling and simulation and the impact brough to the discussion is difficult to ascertain since it is almost totally absent from the manuscript. From the computational details it seams that the authors performed potential energy surface scan along the K/imidazole distance however such curves are not presented. It would be important to show them to visuyalize the different crossing between the states and helping understanding from the one side the stabilization of the parent compound and from the other the competition between diabatic and adiabatic processes. Indeed, those results should be used to provide a reasonable support to the discussion and the conclusion drawn, at present the Table containing just the natural orbitals is of almost no utility to that purpose, and indeed it is scarcerly cited in the manuscript. 

In a nutshell, either the author properly present and discuss modeling and simulation data or they remove them entirely. 

In addition some details concerning the computational methodology, such as the chosen active space should be provided, as well as a discussion on the effects of neglecting dynamic correlation. 

Some other sessions of the manuscript should be revised, in particular the discussion between adiabatic and diabatic transitions is quite convoluted and should profit from a consisting rephrasing. 

A reorganization of the figures could also be considered to enhance readability. 

Finally when citing the effects of OH radical and reactive oxygen species on DNA in the framework of radiosensitization the authors could consider citing the work of Cadet and Ravanat as well as some reviews of Dumont and Monari on the subject. 

Author Response

In particular, the role of molecular modeling and simulation and the impact brought to the discussion is difficult to ascertain since it is almost totally absent from the manuscript.

 Authors’ reply: although this is mostly experimental work, the authors totally agree with the reviewer’s comment. Considering this, we have added a fully comprehensive discussion throughout the manuscript taking into account the calculation results for the different anionic species observed. Thus, we have added the following paragraphs to the different sections as:

 Section 4.1.1:

“A close inspection of Table 1 electron spin densities reveals that the LUMOs of the nitroimidazole compounds show that electron promotion may be localized in the five-member ring. However, due to nitrogen dioxide high electron affinity (2.2730 ± 0.0050 eV) [38] a reasonable delocalization of such LUMOs are expected over NO2, which is not visible due to the truncation methodology employed plotting those MOs. Thus, parent ion formation can only be rationalized in terms of strong competition between the ring and NO2 upon electron transfer. Actually, the branching rations in Figures 7, 8 and 9 clearly show either the parent anion or NO2- formation, such processes competing as a function of the collision energy. The yield of such anions is also dependent on collision time and the efficient mechanism of intramolecular electron transfer from the ring to NO2 and vice-versa, which may be strongly related with the stabilizing effect of K+ in the vicinity of the TNI. Additionally, the calculated LUMOs in Table 1 also show the different character of 4(5)NI, 1Me4NI and Me5NI, σ*(C=C), relative to the π*(ring) of 2NI. Given that in principle an electron promotion to a π* orbital does not lead to bond breaking, unless a σ* is diabatically crossed with such MO, parent ion formation relative to NO2in 2NI is expected to be more enhanced than in 4(5)NI, Me4NI and Me5NI. Such behaviour is clearly visible in the relative intensities of such anions in the TOF mass spectra of Figure 4. In this collision energy regime (~16 eV available energy), accessing the lowest MOs seems plausible, e.g. in 2NI π* ring at 4.9 eV, whereas at intermediate energies (see Figure 5) accessing higher energy π* orbitals (6.1 and/or 7.4 eV) is possible as well as an efficient crossing with σ* antibonding orbital (9.0 eV). Then this will yield an enhanced fragmentation pattern where the ratio between parent anion and NO2 is lower as is the case of 2-nitroimidazole.”

 Section 4.1.2:

“From the calculated MOs in Table 1, generally speaking, transitions from the HOMOs to the LUMOs reveal that the overlap is more efficient in the case of 4(5)NI, Me4NI and Me5NI rather than in 2NI, which means an enhanced yield of NO2 for the former set of molecules than the latter.”

 Section 4.1.4:

“The cyanide anion yields in Figure 4 are discernible although with modest intensity for 2NI, 4(5)NI and Me4NI while in the case of Me5NI is completely suppressed. Such is quite interesting because the calculated MOs in Table 1 show similar electron spin densities. However, at intermediate collision energies (Figure 5), CN‒ formation is possible for all investigated molecules but is more enhanced in the case of 2NI. Such is certainly expected from the considerable π* ring character of the LUMO at 4.9 eV and even at 6.1 eV, that can be accessed at these energies.”

 Section 4.2.2:

“Figure 10 and 11 show CN as one of the most relevant anions formed in K−imidazole collisions, which seems reasonable given the electron affinity of the cyano radical (3.8620 ± 0.0050 eV) [50]. Indeed, it is the second most intense fragment anion between 6.5 and 35.7 eV (19 eV and 70 eV lab frame). Yet, the relative intensity of this anion to the dehydrogenated parent anion at ~13 eV can be explained by initial electron transfer to the lowest MO (see Table 1), that is mainly of π* ring character, and so, not particularly efficient yielding CN. We observe that CN threshold of formation is 6.5 eV (19 eV lab frame). For higher energies, especially for 167 eV (300 eV lab frame), we also observe a strong competition between CN and C2H2N formation. The electron affinity of the cyanomethyl radical is (1.530 ± 0.010 eV) [50] which may indicate that the imidazole collision induced   dissociation process is mainly dictated (> 53 eV, see Figure 10) by the excess energy that is deposited in IMI internal degrees of freedom (...)”

From the computational details it seems that the authors performed potential energy surface scan along the K/imidazole distance however such curves are not presented. It would be important to show them to visualize the different crossing between the states and helping understanding from the one side the stabilization of the parent compound and from the other the competition between diabatic and adiabatic processes. Indeed, those results should be used to provide a reasonable support to the discussion and the conclusion drawn, at present the Table containing just the natural orbitals is of almost no utility to that purpose, and indeed it is scarcely cited in the manuscript. In a nutshell, either the author properly present and discuss modeling and simulation data or they remove them entirely. 

 Authors’ reply: potential energy surface scans were not performed and are not envisaged due to the manuscript’s scope as well as the quite extensive length of it. However, in order to clarify the readers about the impact of theoretical calculations in the discussion of experimental results we have re-written parts of the manuscript.

In addition, some details concerning the computational methodology, such as the chosen active space should be provided, as well as a discussion on the effects of neglecting dynamic correlation. 

 Authors’ reply: The active space has been detailed in the section Theoretical method. The calculation takes into account only the static electron correlation which provides a reasonable description of the molecular orbitals.

‘The natural molecular orbitals of the imidazole compounds in presence of potassium have been determined in the asymptotic region at the state-averaged CASSCF level of theory from a CAS(3,11) calculation taking account of the static electron correlation as in previous work [33] and involving three active electrons and eleven active orbitals including molecular orbitals on imidazole compounds and s,p orbitals on potassium.’

 Some other sessions of the manuscript should be revised, in particular the discussion between adiabatic and diabatic transitions is quite convoluted and should profit from a consisting rephrasing.

 Authors’ reply: in order to make the discussion about the adiabatic and diabatic transitions more clear some parts of the text were modified and some text was added as well as two new figures (2 and 3). The following text was introduced in Section 4:

“In atom-molecule collisions where an electron transfer occurs, a negative ion is formed as part of intermediate step or as final product. The electron transfer process happens when electrons follow adiabatically the nuclear motion in the vicinity of the crossing of the stationary nonperturbed states, i.e. the covalent and the ionic diabatic states [37] (see Figure 2). Considering a case of a diatomic molecules, the ionic surface lies above the covalent surface where the endoergicity (∆E) at large atom-molecule distances is given by ∆E = IE(K) ‒ EA(AB), being IE the ionization energy of potassium atom and EA the electron affinity of the target molecule.”

 A reorganization of the figures could also be considered to enhance readability. 

 Authors’ reply: figures were reorganized as requested.

Finally, when citing the effects of OH radical and reactive oxygen species on DNA in the framework of radiosensitization the authors could consider citing the work of Cadet and Ravanat as well as some reviews of Dumont and Monari on the subject. 

 Authors’ reply: the following publications were added to the manuscript, namely in section 4.1.3:

- Cadet, J.; Douki, T.; Ravanat, J.L. Oxidatively generated base damage to cellular DNA. Free Radic. Biol. Med. 2010, 49, 9–21.

- Cadet, J.; Douki, T.; Ravanat, J.L. Measurement of oxidatively generated base damage in cellular DNA. Mutat. Res. - Fundam. Mol. Mech. Mutagen. 2011, 711, 3–12.

- Ravanat, J.-L.; Cadet, J.; Douki, T. Oxidatively Generated DNA Lesions as Potential Biomarkers of In Vivo Oxidative Stress. Curr. Mol. Med. 2012, 12, 655–671.

- Dumont, E.; Monari, A. Understanding DNA under oxidative stress and sensitization: The role of molecular modeling. Front. Chem. 2015, 3, 1–15.

You can also see the attachment.

Reviewer 2 Report

Referee’s comments

Title: Electron-Transfer-Induced Decomposition of Nitroimidazoles: Experimental and Theoretical Methods

Authors : M Mendes et al.

This paper has reported electron transfer induced decomposition of Nitroimidazoles etc. due to the collision of K atoms. I think that this study is important in the field of molecular science and useful for the development of radio-therapeutic drug and treatments. Therefore. I recommend this paper to be published in International Journal of Molecular Science after the revisions of following points.

Title:

The content of ’collision with K atoms’ should be reflected in the title. When I see this title, I wonder how electron transfer is induced.

Section 3 Theoretical method (page 3):

In the first of this section, you should mention what you calculate concretely such as the structure and the biding energies of initial, final, and intermediate states. I guess that the calculations are useful to produce Eq.(1)-(4) and Table 1, however, it is only the guess.

Section 4.1.1 Parent anion formation (page 5):

It is very difficult to understand contents of this section. Although four or five different stories are included, this section is composed of one paragraph. Whenever the story changes, you should force a line break. Line 23-25: ‘However, ----'

I guess that the reason of ‘quite interesting’ is ‘fast collisions follow the adiabatic principle and parent anions are not produced in the adiabatic principle, however, parent anions are produced in fast collisions’. I do not know if this guess is correct or not. However, in order to understand what you mention, such the guess is required. You should rewrite these sentences.

‘It seems to not follow’ should be changed ‘It seems not to follow’?

Line 30:

You should explain ‘the approximation’ in more detail.

Lines 32-33 : the different degree of ---

You should explain what ‘autodetachment’ is using chemical reaction formulas and why the autodetachment is delayed due to the excess of the internal energy.

Line 50

What is ‘a third body’. You should explain it by chemical formulas.

Section 4.1.2 Formation of NO2----- (page 8)

Line 4 :

I cannot see (M-NO2)-in 4(5)NI in Figs.2 and 4. Is this contradictory to the sentence of this line?

Author Response

Title: The content of ’collision with K atoms’ should be reflected in the title. When I see this title, I wonder how electron transfer is induced.

 Authors’ reply: changed as per in the reviewer comment.

 Section 3 Theoretical method (page 3):

In the first of this section, you should mention what you calculate concretely such as the structure and the biding energies of initial, final, and intermediate states. I guess that the calculations are useful to produce Eq.(1)-(4) and Table 1, however, it is only the guess.

 Authors’ reply: Calculations of the K+nitroimidazole system are performed in the asymptotic region in order to describe the molecular orbitals involved in the collision of K on nitroimidazole. Molecular states are constructed on these MOs and their energies may be determined. These MOs are displayed in Table 1.

Section 4.1.1 Parent anion formation (page 5):

It is very difficult to understand contents of this section. Although four or five different stories are included, this section is composed of one paragraph. Whenever the story changes, you should force a line break. Line 23-25: ‘However, --'

 Authors’ reply: corrected.

 I guess that the reason of ‘quite interesting’ is ‘fast collisions follow the adiabatic principle and parent anions are not produced in the adiabatic principle, however, parent anions are produced in fast collisions’. I do not know if this guess is correct or not. However, in order to understand what you mention, such the guess is required. You should rewrite these sentences.

 Authors’ reply: ‘interesting’ has been removed from the different citations and changed accordingly.

 ‘It seems to not follow’ should be changed ‘It seems not to follow’?

 Authors’ reply: changed.

 Line 30:

You should explain ‘the approximation’ in more detail.

 Authors’ reply: ‘approach’ is much more attuned to the description. This has been changed.

 Lines 32-33 : the different degree of ---

You should explain what ‘autodetachment’ is using chemical reaction formulas and why the autodetachment is delayed due to the excess of the internal energy.

 Authors’ reply: autodetachment is related to the ejection from the TNI of the transferred electron. If the excess energy due to the electron transfer process is redistributed through the different degrees of freedom of the TNI, then autodetachment probability decreases allowing a stable negative ion to be formed.

 Line 50

What is ‘a third body’. You should explain it by chemical formulas.

 Authors’ reply: in the sentence, added in brackets as (K+). In this sort of collisions apart from the target molecule (M) and the electron (e-) being transferred, a positive potassium cation (K+) is formed. As so in the vicinity of the temporary negative ion (M-), the two body system, K+ is considered a third body.

 Section 4.1.2 Formation of NO2----- (page 8)

Line 4 :

I cannot see (M-NO2)-in 4(5)NI in Figs.2 and 4. Is this contradictory to the sentence of this line?

 Authors’ reply: although the signal is very weak, authors have decided not to include it to avoid congesting the figure. In any case the sentence has been changed accordingly.

You can also see the attachment.

Reviewer 3 Report

The paper "Electron-Transfer-Induced Decomposition of Nitroimidazoles: Experimental and Theoretical Methods" by M. Mendes, G. García, M.-C. Bacchus-Montabonel and P. Limão-Vieira features a comparative study of electron attachment to imidazole and several nitroimidazoles. The paper presents new original results, is well written and deserves publication in the International Journal of Molecular Sciences. Still, I would suggest that the authors consider the issues below before preparing the final version of their manuscript:

1) Page 5, discussion between lines ~23-50: The discussion is hard to follow for a person not working in the field. I would suggest adding a Figure to make the points clear, mainly with respect to the following part: "...potassium atom may not transfer its electron at the first crossing but rather at the second."

2) The authors claim that they see CN- in the experiment. Is their resolution good enough to rule out that C2H2- is formed?

3) Page 9: The word "extremely" is used twice (in its first occurrence, it should be rather "extreme"). The authors might consider using a different, less emotional word.

4) Table 2: The authors should revise the table, as some energies cannot be seen at all in my .pdf version. Also, I believe that 9.3 eV in the first row should read -9.3 eV.

5) A supporting information file with all structures used for quantum chemical calculations seems to be missing.

Typos/formulations:

1) Page 1, line 34: "nitro" -> "nitro group"

2) In chemical equations, a dot to denote radicals is only sometimes used. The authors should unify the style.

3) Page 11: In (IMI−H)−, the minus should be in superscript.

Author Response

1) Page 5, discussion between lines ~23-50: The discussion is hard to follow for a person not working in the field. I would suggest adding a Figure to make the points clear, mainly with respect to the following part: "...potassium atom may not transfer its electron at the first crossing but rather at the second."

 Authors’ reply: two new figures (2 and 3) were added.

 2) The authors claim that they see CN- in the experiment. Is their resolution good enough to rule out that C2H2- is formed?

 Authors’ reply: the mass resolution in these experiments was around 700. Since the mass of CN is 26.0174 u and the C2H2 is 26.0373 u it would be necessary a mass resolution of 1306 to completely distinguish both molecular species.

 3) Page 9: The word "extremely" is used twice (in its first occurrence, it should be rather "extreme"). The authors might consider using a different, less emotional word.

 Authors’ reply: corrected. The expression “seem to be of extremely” was replace by “are quite important”. In the second occurrence the word “extremely” was replaced by “particularly”.

 4) Table 2: The authors should revise the table, as some energies cannot be seen at all in my .pdf version. Also, I believe that 9.3 eV in the first row should read -9.3 eV.

 Authors’ reply: Table 1 (before Table 2) was revised and modified in order to avoid this changes when converted in .pdf.

 5) A supporting information file with all structures used for quantum chemical calculations seems to be missing.

 Authors’ reply: a file for supporting material was created including the molecules structures used in the calculations and it will be submitted with the re-submission of the manuscript.

 Typos/formulations:

 1) Page 1, line 34: "nitro" -> "nitro group"

 Authors’ reply: this was corrected.

 2) In chemical equations, a dot to denote radicals is only sometimes used. The authors should unify the style.

 Authors’ reply: this was corrected.

 3) Page 11: In (IMI−H)−, the minus should be in superscript.

 Authors’ reply: this was corrected.

You can also see the attachment.

This manuscript is a resubmission of an earlier submission. The following is a list of the peer review reports and author responses from that submission.